# Peroxisome Proliferator-Activated Receptor-Targeted Therapies: Challenges upon Infectious Diseases

**DOI:** 10.3390/cells12040650

**Published:** 2023-02-17

**Authors:** In Soo Kim, Prashanta Silwal, Eun-Kyeong Jo

**Affiliations:** 1Department of Microbiology, College of Medicine, Chungnam National University, Daejeon 35015, Republic of Korea; 2Department of Medical Science, College of Medicine, Chungnam National University, Daejeon 35015, Republic of Korea; 3Infection Control Convergence Research Center, College of Medicine, Chungnam National University, Daejeon 35015, Republic of Korea; 4Department of Orthopaedic Surgery, University of Pittsburgh School of Medicine, Pittsburgh, PA 15261, USA

**Keywords:** peroxisome proliferator-activated receptor, infection, bacteria, virus, parasite

## Abstract

Peroxisome proliferator-activated receptors (PPARs) α, β, and γ are nuclear receptors that orchestrate the transcriptional regulation of genes involved in a variety of biological responses, such as energy metabolism and homeostasis, regulation of inflammation, cellular development, and differentiation. The many roles played by the PPAR signaling pathways indicate that PPARs may be useful targets for various human diseases, including metabolic and inflammatory conditions and tumors. Accumulating evidence suggests that each PPAR plays prominent but different roles in viral, bacterial, and parasitic infectious disease development. In this review, we discuss recent PPAR research works that are focused on how PPARs control various infections and immune responses. In addition, we describe the current and potential therapeutic uses of PPAR agonists/antagonists in the context of infectious diseases. A more comprehensive understanding of the roles played by PPARs in terms of host-pathogen interactions will yield potential adjunctive personalized therapies employing PPAR-modulating agents.

## 1. Introduction

Peroxisome proliferator-activated receptors (PPARs) are adopted orphan family members of the nuclear receptor group that regulates various biological functions, including glucose and lipid homeostasis, inflammation, and adipose cell differentiation [1,2]. PPARs are ligand-activated transcription factors that are subdivided into three isoforms, termed PPARα (NR1C1), PPARβ/δ (also termed PPARβ or PPARδ, or NR1C2), and PPARγ (NR1C3) [3]. The endogenous PPAR ligands include long-chain polyunsaturated fatty acids and eicosanoids, although the functions of the ligands remain largely unknown [2,4]. Each PPAR isoform evidences a distinct cellular and tissue distribution and biological functions with a focus on energy balance and inflammation [2].

PPARs feature N-terminal DNA-binding and C-terminal ligand-binding domains and form heterodimers with nuclear retinoid X receptor (RXR)-α [5,6]. After interacting with the ligands, PPAR-RXR heterodimers undergo conformational changes that allow them to regulate the transcription of many genes with peroxisome proliferator response elements (PPREs) in their promoter regions [7]. The many PPAR-mediated functions are orchestrated via recruitment of different transcriptional co-activators, including PPAR co-activator-1α, co-activator-associated proteins, and co-repressors [2,5]. Moreover, each PPAR isoform transcriptionally regulates the expression of the other PPAR isoforms via feedback control [2].

PPARα is found principally in the liver and transcriptionally regulates fatty acid oxidation, cholesterol and glycogen metabolism, gluconeogenesis, ketogenesis, and inflammation [8,9]. PPARγ is found in both hematopoietic and non-hematopoietic cells and tissues (adipose tissue and the large intestine) [10]. PPARγ modulates many biological functions, including fatty acid and glucose metabolism and anti-inflammatory signaling via nuclear factor kappa B (NF-κB); it also suppresses oxidative stress and prevents platelet-leukocyte interactions [10,11]. Recent insights into the roles played by PPAR ligands have enabled development of PPAR agonists/antagonists, which serve as candidate drugs for inflammatory, metabolic, and autoimmune diseases, as well as cancers [12]. Several PPARα ligands, including fibrates, helpfully treat dyslipidemia, while the PPARγ ligands pioglitazone and rosiglitazone are well-known anti-diabetic drugs [13]. The three PPARs play critical but distinct roles in regulating the inflammation and metabolic pathways closely associated with immune cell functions [14,15,16]. It is thus essential to understand how PPARs affect antimicrobial actions against diverse infections. Here, we highlight recent insights into how the PPAR isoforms and their agonists regulate antimicrobial host defenses against viral, bacterial, and parasitic diseases.

## 2. Overview of PPARs

### 2.1. Molecular Characteristics of PPARs 

Peroxisomes, 0.5 μm diameter single-membrane cytoplasmic organelles, play essential roles in the oxidation of various biomolecules [17,18]. Peroxisome proliferators are multiple chemicals that increase the abundance of peroxisomes in cells [19,20]. These molecules also increase gene expression for β-oxidation of long-chain fatty acids and cytochrome P450 (CYP450) [21,22]. Given the gene transcriptional modulation of peroxisome proliferators, PPARs have been identified as nuclear receptors [23,24,25,26,27,28,29]. The PPAR subfamily consists of three isoforms, PPARα, PPARβ/δ, and PPARγ [30]. The three PPARs differ in tissue-specific expression patterns and ligand-biding domains, each performing distinct functions. *PPARA*, encoding PPARα, is located in chromosome 22q13.31 in humans and is mainly expressed in the liver, intestine, kidney, heart, and muscle [31,32]. PPARγ has four alternative splicing forms from *PPARG* located in chromosome 3p25.2 and is highly expressed in adipose tissue, the spleen, and intestine [33,34]. PPARδ, encoded by *PPARD*, is located in chromosome 6q21.31 and presents ubiquitously [29,35]. Thus, it is essential in the study of PPARs to consider their tissue distribution and functions. 

PPAR is a nuclear receptor superfamily class II member that heterodimerizes with RXR [36,37]. The PPAR structure includes the A/B, C, D, and E domains from N-terminus to C-terminus [38]. The N-terminal A/B domain (NTD) is a ligand-independent transactivation domain containing the activator function (AF)-1 region. The NTD is targeted for variable post-translational modifications, including SUMOylation, phosphorylation, acetylation, O-GlcNAcylation, and ubiquitination, resulting in transcriptional regulatory activities [39]. DNA-binding C domain (DBD) has two DNA-binding zinc finger motifs containing cysteines, which dock to PPREs. PPARs reside upstream of RXR upon the direct repeat (DR)-1 motifs, which are composed of two hexanucleotide consensus sequences with one spacing nucleotide (AGGTCA N AGGTCA) [40]. The hinge D region is a linker between the C and E domains, which contains a nuclear localization signal, and is the site for post-translational modifications such as phosphorylation, acetylation, and SUMOylation [39]. The ligand-binding E domain (LBD) carries the hydrophobic ligand-binding pocket and the AF-2 region. The absence of agonists enables LBD to recruit co-repressors containing the CoRNR motifs [41]. Engaging agonists to LBD elicits conformational changes of AF-2 to facilitate interaction with LXXLL motifs of many co-activators [42]. Like other nuclear receptor superfamily class II members, such as thyroid hormone receptor (TR), retinoic acid receptor (RAR), and vitamin D receptor (VDR), PPARs function as heterodimers with RXR through LBD [6,43]. LBD is also targeted for SUMOylation and ubiquitination [39]. Advancement of research on the PPAR structure helps thoroughly dissect the roles of PPARs. We will discuss the roles of specific PPAR subtypes in the following subsections.

#### 2.1.1. Roles of PPARα

PPARα is predominantly expressed in the liver but is also found in other tissues, including the heart, muscle, and kidney [4,32]. PPARα regulates the expression of genes involved in metabolism and inflammation. Activation of PPARα leads to the upregulation of genes involved in fatty acid oxidation and the downregulation of genes involved in fatty acid synthesis [8]. PPARα also modulates other genes, including genes involved in the transport and uptake of fatty acids and the synthesis and secretion of lipoproteins [4,8]. In addition, activation of PPARα has been shown to improve insulin sensitivity, reduce oxidative stress, and reduce inflammation in preclinical studies [7,8,44]. PPARα activation has been shown to modify the expression of immune response genes, including those encoding cytokines and chemokines, which are signaling molecules that regulate the immune response [44,45]. PPARα activation has also been demonstrated to reduce the production of pro-inflammatory cytokines, such as tumor necrosis factor-alpha (TNF-α) and interleukin (IL)-6 [46,47]. PPARα has been shown to interfere with the DNA binding of both AP-1 and NF-κB [45,46,48]. Thus, the roles of PPARα in infectious diseases should be studied in wide ranging aspects, including metabolism and inflammation.

In the context of infection, PPARα has been shown to play an essential role in the hepatic metabolic response to infection. During an infectious challenge, the liver coordinates several metabolic changes to support the host defense response, including the mobilization of energy stores, production of acute-phase proteins, and synthesis of new metabolites. Activation of PPARα in the liver leads to the upregulation of genes involved in fatty acid oxidation and ketogenesis with fibroblast growth factor 21 (FGF21) production [49]. FGF21 is a hormone produced by the liver that has been shown to promote ketogenesis and reduce glucose utilization [50,51]. The ketogenesis regulation of PPARα with FGF21 is essential for reacting to microbial or viral sepsis [52,53,54]. In conclusion, the hepatic PPARα metabolic response to infection is crucial to the host defense response.

#### 2.1.2. Roles of PPARβ/δ

PPARβ is expressed in diverse tissues, including adipose tissue, muscle, and the liver [29,55], and is activated by multiple ligands, such as fatty acids and their derivatives [7]. PPARβ is involved in regulating lipid metabolism and energy homeostasis, as well as controlling inflammation and immune function [56]. PPARβ activation has been demonstrated to have pro- and anti-inflammatory effects based on the situation [56]. The role of PPARβ in tumorigenesis is debatable. PPARβ activation has been found in some cases to have anti-tumorigenic effects, such as causing apoptosis and inhibiting cell proliferation [57,58]. In other cases, however, activation of PPARβ has been shown to promote tumorigenesis by enhancing cell survival, promoting angiogenesis, and reducing cellular differentiation [59,60,61,62]. Overall, the role of PPARβ activation in cancer is not entirely known and is complex. Similarly, the function of PPARβ in infection is not well understood. Additional research is required to comprehend the function of PPARβ in the context of immunology against cancers and infectious diseases.

#### 2.1.3. Roles of PPARγ

PPARγ is expressed in a variety of tissues, including adipose tissue, muscle, and the liver [33,34,55], and is activated by diverse ligands, including fatty acids and their derivatives, as well as synthetic chemicals known as thiazolidinediones [4,7]. PPARγ is responsible for regulating lipid metabolism, glucose homeostasis, and inflammation [63,64]. Numerous inflammatory mediators and cytokines are inhibited by PPARγ ligands in various cell types, including monocytes/macrophages, epithelial cells, smooth muscle cells, endothelial cells, dendritic cells, and lymphocytes. In addition, PPARγ diminishes the activities of transcription factors AP-1, STAT, NF-κB, and NFAT to adversely regulate inflammatory gene expression [65,66,67]. As a result, PPARγ has been demonstrated to have a protective function against infections by modulating the immune response and lowering inflammation. However, other researchers have hypothesized that PPARγ activation may impair the function of immune cells, such as macrophages, and contribute to the development of infections. Therefore, the role of PPARγ in disease is complex and context-dependent, and more research is needed to fully understand the molecular mechanisms by which PPARγ regulates the host response to infection.

### 2.2. Regulatory Mechanisms of PPARs

The PPAR ligand-binding pocket is large and capable of engaging diverse ligands [68,69]. Endogenous ligands vary depending on the PPAR isoform, including n-3 polyunsaturated fatty acids such as docosahexaenoic acid and eicosapentaenoic acid for all PPARs, leukotriene B4 for PPARα, carbaprostacyclin for PPARδ, and prostaglandin J2 for PPARγ [70]. Representative synthetic agonists include fibrates (PPARα agonists) and thiazolidinediones (PPARγ agonists) [7]. Fibrates, such as fenofibrate, clofibrate, and gemfibrozil, are widely used for treating dyslipidemia. Thiazolidinediones, such as rosiglitazone, pioglitazone, and lobeglitazone, improve insulin resistance [7]. Most clinical studies on PPAR actions in infectious diseases have been conducted retrospectively, and no clinical studies currently in progress are listed in ClinicalTrials.gov (https://clinicaltrials.gov/ (accessed on 13 February 2023)). Since widely used PPAR agonists exist, clinical research can be conducted through a deeper understanding of PPAR roles in infectious diseases.

PPAR-RXR heterodimerization occurs ligand-independently [6]. The heterodimer appears to exert transcriptional regulation both ligand-dependently and -independently [7]. Although LBD may interact with either co-repressor or co-activator in the state of not binding with an agonist, binding to a ligand elicits stabilized co-activator-LBD interaction, thus increasing transactivation [7,71]. Further, recent studies have shown that PPARs inhibit other transcription factors, such as NF-κB, activator protein-1 (AP-1), signal transducer and activator of transcription (STAT), and nuclear factor of activated T cells (NFAT) [44,65,66,67]. Recent studies revealed the possibility of forming a protein chaperone complex with PPAR-associated proteins, such as heat shock proteins (HSPs). Similar to interactions between other type I intracellular receptors and heat shock proteins, HSP90 repressed PPARα and PPARβ activities but not that of PPARγ [72]. Instead, HSP90 was required for PPARγ signaling in the nonalcoholic fatty liver disease mouse model [73]. Thus, it is necessary to study the various modes of PPAR actions. The intracellular regulatory mechanisms of PPARs are shown in Figure 1.

## 3. PPARs and Viral Infections

### 3.1. PPARs and Respiratory Viral Infections

Many studies have shown that PPARγ controls viral replication and virus-associated inflammation by antagonizing inflammatory signaling pathways such as the NF-κB and STAT pathways [74,75]. In particular, PPARγ of alveolar macrophages critically modulates acute inflammation to promote recovery from respiratory viral infections, most of which are caused by influenza A virus (IAV) and respiratory syncytial virus (RSV) [76]. Several PPAR agonists have shown promise in terms of ameliorating virus-related cytokine storms and the damage caused by severe IAV infection [77]. Macrophage PPARγ is essential for resolving the chronic pulmonary collagen deposition and fibrotic changes that follow influenza infection [78]. Several researchers have sought new therapeutic candidates for IAV disease. A recent screening of traditional Chinese medicines showed that emodin and analogs thereof evidenced excellent anti-IAV activities mediated by activation of the PPARα/γ and adenosine monophosphate (AMP)-activated protein kinase (AMPK) pathways [79]. High-throughput screening of natural compounds and/or synthetic drugs/agents will yield new therapeutics against respiratory viral infections based on drug interactions with PPAR pathways. 

A link has been suggested between severe acute respiratory syndrome coronavirus 2 (SARS-CoV-2) virus infection and PPARα activity in the context of lipid uptake, lipotoxicity, and vascular inflammation [80,81,82]. The PPARα agonist fenofibrate is a potential adjunctive coronavirus disease (COVID-19) therapy; the material exhibits anti-inflammatory and anti-thrombotic activities [80,82]. A study employing a public database on subjects with type 2 diabetes and COVID-19, along with animal studies, revealed that the PPARγ agonist pioglitazone may ameliorate acute lung injury and SARS-CoV-2-mediated hyperinflammation [83]. Cannabidiol working via PPARγ is proposed as a therapeutic approach for the severe form of COVID-19 [84]. A recent study demonstrated that cannabidiol attenuated inflammation and epithelial damage in colonic epithelial cells exposed to the SARS-CoV-2 spike protein through a PPARγ-dependent mechanism [85]. The natural compound γ-oryzanol may also serve as an adjunctive therapy to reduce the cytokine storm associated with COVID-19; the material stimulated PPARγ to modulate oxidative stress and the inflammatory response in adipose tissues [86]. The Middle East respiratory syndrome coronavirus (MERS-CoV)-derived S glycoprotein activates PPARγ to suppress the pathologic inflammatory responses of macrophages [87]. Further research on the modulatory roles played by PPAR agonists/antagonists in terms of virus-associated inflammation will yield novel adjunctive therapeutics to counter emerging and re-emerging viral infections. Table 1 summarizes studies on PPARs and their ligands in relation to viral infections.

### 3.2. PPARs and Virus-Related Inflammation

A recent study showed that the inflammatory responses during infection with Chikungunya virus (CHIKV) involved the renin-angiotensin system (RAS) and PPARγ pathways [88]. The telmisartan-mediated suppression of CHIKV infection is at least partly mediated via activation of PPARγ; a PPARγ antagonist increased the CHIKV viral load [88]. Omeragic et al. showed that PPARγ played a critical role in terms of human immunodeficiency virus (HIV-1) ADA glycoprotein 120 (gp120)-related inflammatory marker generation was observed in primary astrocytes and microglia and also in vivo [89]. The anti-inflammatory activities induced by the PPARγ agonists rosiglitazone and pioglitazone reflected suppression of the NF-κB signaling pathway [89]. These relationships between PPARγ and viral infections are included in Table 1. Δ-9-tetrahydrocannabinol improved epithelial barrier function and thus protected colonic tissues of rhesus macaques chronically infected with simian immunodeficiency virus (SIV). This was at least partly attributable to the upregulation of PPARγ [92]. PPARα signaling is required for restoration of the intestinal barrier by the probiotic *Lactobacillus plantarum* and amelioration of gut inflammation during SIV infection [93]. Such findings strongly suggest that targeting PPARγ would both prevent and treat virus-associated inflammation of the brain, endothelial system, and intestinal tissues. The PPARγ antagonist GW9662 protected against dengue virus infection and di(2-ethylhexyl) phthalate (DEHP)-induced interleukin (IL)-23 expression, thus suppressing the viral load [94]. Therefore, future clinical trials should explore the protective effects of several possible PPAR agonists/antagonists and combinations thereof with current antivirals in patients with various viral infections.

Zika virus (ZIKV) is a serious arthropod-borne (arbovirus) pathogen that causes congenital defects and neurological diseases in both infants and adults [95]. A recent study showed that ZIKV-induced cellular responses of induced pluripotent stem cell (iPSC)-derived neural progenitor cells involved the PPAR signaling pathways, which may contribute to neurogenesis and viral replication [96]. However, further research is required.

### 3.3. PPARs and Hepatitis Virus Infection

The roles played by PPAR pathways in terms of hepatitis B virus (HBV) infection elimination are complex. IL-1β production induced by HBV infection of M1-like inflammatory macrophages triggered anti-HBV responses via downregulation of PPARα and forkhead box O3 (FOXO3) expression in hepatocytes [97]. OSS_128167, a sirtuin 6 inhibitor, inhibited HBV transcription and replication in hepatic cells and in vivo by targeting PPARα expression [90]. In the HBV replicative mouse model, PPAR agonists, including bezafibrate, fenofibrate, and rosiglitazone, significantly increased the serum levels of HBV antigens HBsAg, HBeAg, and HBcAg and that of HBV DNA, as well as the viral load in mouse liver [98]. Thus, patients with metabolic diseases taking PPAR-based therapeutics should take care to avoid HBV infection. However, in a retrospective study of HBV-infected patients treated with entecavir and tenofovir-disoproxil-fumarate, the drugs exerted profound extrahepatic effects on lipid metabolism, reducing serum cholesterol levels by inducing the expression of PPARα target genes such as CD36 in liver tissue and cells [99]. Thus, the PPARα-activating nucleoside analogs tenofovir-disoproxil-fumarate may usefully treat atherosclerosis and hepatocarcinogenesis, both of which are associated with dyslipidemia. This would be a new role for an anti-HBV therapeutic. However, the precise functions of PPARs during HBV infection remain unclear. The antiviral, antitumor, and extrahepatic actions of PPAR agonists vary with the clinical condition. 

During hepatitis C virus (HCV) infection, PPAR-α/β/γ stimulators/agonists reduce calcitriol-mediated anti-HCV responses, presumably by counteracting the calcitriol-mediated activation of vitamin D receptor signaling and inhibiting nitrative stress [91]. Naringenin, a grapefruit flavonoid, suppressed HCV production by inhibiting viral particle assembly via PPARα activation, suggesting potential roles for PPARα agonists in the resolution of infection [100]. It is essential to perform an in-depth exploration of how the three PPARs and their signaling pathways affect the outcomes of HBV and HCV infections. Studies on PPARs and hepatitis virus infections are summarized in Table 1.

## 4. PPARs and Bacterial Infections

### 4.1. PPARs and Post-Influenza Bacterial Infections 

PPARs exacerbate the severity of post-influenza bacterial infections. During *Staphylococcus aureus* superinfection following IAV infection, the levels of CYP450 metabolites, which are PPARα ligands, increase significantly and trigger receptor-interacting serine/threonine-protein kinase 3 (RIPK3)-induced necroptosis, thus exacerbating the lung pathology and increasing mortality from secondary bacterial infection [101]. The PPARγ agonist rosiglitazone reduces bacterial clearance during secondary bacterial pneumonia, which is a frequent complication of primary IAV infection [102]. Diabetic patients treated with rosiglitazone exhibited increased mortality from IAV-associated pneumonia compared to those not treated with rosiglitazone, as revealed by data from the National Health and Nutrition Examination Survey (NHANES) [102]. CYP450 metabolites reduced the protective inflammatory responses via PPARα activation, thereby increasing the susceptibility to secondary bacterial infection following IAV infection [103]. Thus, PPARα or PPARγ drives host protection but reduces bacterial clearance at different stages of IAV infection. The molecular mechanisms by which PPARα/γ mediates immune modulation during a bacterial infection following IAV infection require urgent attention. Better medicines are needed to treat the different stages of IAV-associated disease, which is often fatal in susceptible patients. 

### 4.2. PPARs in Bacterial Infections 

PPARs and agonists/antagonists thereof may modulate disease severity and outcomes in patients with bacterial infections and associated inflammation. In a model of intestinal colitis, 5-aminosalicylic acid, a PPARγ agonist, exerted therapeutic anti-inflammatory effects by activating the epithelial PPARγ signaling pathway [104]. After infection with *Klebsiella pneumoniae*, which is the respiratory Gram-negative bacterium that usually causes pneumonia, PPARγ agonists such as pioglitazone reduced proinflammatory cytokine and myeloperoxidase levels, bacterial growth in lung tissues, and bacterial dissemination to distant organs [105]. The taste receptor type-2 member 138 (TAS2R138) plays a role in neutrophil-associated host innate immune defense after *Pseudomonas aeruginosa* infection [106]. TAS2R138 mediated the degradation of lipid bodies via competitive binding to the PPARγ antagonist N-(3-oxododecanoyl)-L-homoserine lactone (AHL-12), a mediator of virulence produced by *P. aeruginosa* [106]. Although the exact roles of PPARγ in antimicrobial responses remain unclear, a study employing a model of *P. aeruginosa* infection found that the PPARγ agonist pioglitazone increased the levels of certain chemokines (*Cxcl1*, *Cxcl2*, and *Ccl20*) and cytokines (*Tnfa*, *Il6*, and *Cfs3*) in bronchial epithelial cells and suppressed inflammatory responses in bronchoalveolar lavage fluid [107]. Future studies must explore the utility of PPAR agonists/antagonists as adjuvant therapies and determine whether systemic or local treatments improve disease outcomes. 

During *Chlamydia pneumoniae* infection, both PPARα and PPARγ are required to upregulate foam macrophage formation via induction of the scavenger receptor A1 (SR-A1) and the acyl-coenzyme A cholesterol acyltransferase 1 (ACAT1) involved in cholesterol esterification [108]. PPARα and PPARγ agonists, including fenofibrate and rosiglitazone, may suppress atherosclerotic plaque formation in patients with coronary heart disease infected with *C. pneumoniae* [108]. Activation of both PPARα and PPARγ by PAR5359 protected against *Citrobacter rodentium*-induced colitis. The dual agonism promoted antibacterial immunity and ameliorated the inflammatory response [109].

In contrast to studies with Gram-negative bacteria, few reports have explored the roles played by PPARs during Gram-positive infections. In a *Caenorhabditis elegans* model, induction of the gene encoding flavin-containing monooxygenase (FMO) *fmo-2/FMO5* by NHR-49/PPAR-α was critical in terms of the establishment of an effective innate host defense against *S. aureus* infection [110]. Erythropoietin limits infections caused by Gram-negative *Escherichia coli* and Gram-positive *S. aureus*; macrophage-mediated clearance of these bacteria is at least partly mediated by a PPARγ-dependent pathway [111]. Inhibition of PPARγ signaling reduced the survival of *Rickettsia conorii*, an intracellular Gram-positive bacterium, probably by reducing lipid droplet production [112]. Although PPAR-based therapeutics may counter bacterial infections, more preclinical and clinical studies are required. Table 2 summarizes the roles of PPAR ligands in bacterial infections.

### 4.3. PPARs and Mycobacterial Infections 

Many scholars have sought to clarify the effects of PPARs in those infected with *Mycobacterium tuberculosis* (Mtb) and nontuberculous mycobacteria (NTM), which cause tuberculosis and NTM disease, respectively [113]. Although the relevant bacterial components have not been fully characterized, *M. leprae* and Mtb lead to activation of PPARs [113,114,115]. PPARα and PPARγ appear to play opposite roles. The virulent Mtb strain H37Rv and cell wall component lipoarabinomannan induced PPARγ expression, in turn activating IL-8 and cyclooxygenase (COX) 2 expression, but the attenuated *M. bovis* strain, termed Bacillus Calmette-Guérin (BCG), induced less PPARγ expression [115]. PPARγ activation during Mtb or BCG infection upregulates lipid body formation and increases bacterial survival in macrophages [116,117]. Either PPARγ knockdown or PPARγ antagonist GW9662 increased macrophage-mediated Mtb killing [115,117]. PPARγ activation was associated with enhanced cholesterol and triacylglycerol uptake; these materials are required for macrophage lipid body formation during mycobacterial infection [113]. Antagonists of PPARδ or PPARγ significantly inhibited lipid accumulation by cells infected with *M. leprae*, thus reducing parasitization [114,118]. Together, the data suggest that PPARγ is required for intracellular bacterial survival; PPARγ enhances lipid body formation and foam macrophage development during mycobacterial infection. 

In contrast, PPARα appears to enhance defenses against macrophage and lung Mtb or BCG infection in mice. PPARα-mediated antimicrobial responses are at least partly mediated via promotion of lipid catabolism and activation of the transcription factor EB (TFEB), a transcriptional factor required for lysosomal biogenesis [119]. Notably, PPARα agonists GW7647 and Wy14643 protected macrophages against Mtb or BCG infection [119]. Macrophage PPARα expression reduces inflammatory cytokine synthesis during Mtb or BCG infection [119], suggesting that PPARα ameliorates inflammation. PPARα deficiency reduced the antimicrobial response and increased lung tissue damage during pulmonary *Mycobacteroides abscessus* (Mabc) infection [120]. Gemfibrozil, a PPARα activator, reduced the in vivo Mabc load and lung inflammation during infection [120]. It is important to clarify whether PPARα modulates lipid body formation during infections with Mabc and other NTMs. 

## 5. PPARs and Parasitic Infections

The anti-inflammatory responses of M2 macrophages and Th2 immunity protect against parasitic infections [121]. In allergic patients and those infected with the nematode *Heligmosomoides polygyrus*, PPARγ is highly expressed in Th2 cells. PPARγ affects the development of Th2-associated pathological immune responses and increases IL-33 receptor levels in Th2 cells [122]. *Neospora caninum* infection triggers maturation of M2 macrophage development via upregulation of PPARγ activity and downregulation of NF-κB signaling [123]. In a model of eosinophilic meningoencephalitis caused by the rat lungworm *Angiostrongylus cantonensis*, PPARγ played anti-inflammatory and protective roles by inhibiting NF-κB-mediated pathological inflammatory responses; the PPARγ antagonist GW9662 increased susceptibility to angiostrongyliasis [124]. In a model of cerebral malaria using clinical isolates of *Plasmodium falciparum*, dimethyl fumarate increased the expression of nuclear factor E2-related factor 2 (NRF2), in turn enhancing PPAR signaling and thus ameliorating the neuroinflammatory responses of primary human brain microvascular endothelial cells [125]. Cerebral malaria susceptibility was associated with a lack of PPARγ nuclear translocation and increased COX-2 levels in brain tissues, which was associated with higher-level parasitemia and poorer survival [126]. PPAR signaling may exert useful antiparasitic functions by attenuating inflammation.

*Toxoplasma gondii*, one of the most common zoonotic pathogens, infects both immunocompromised patients and healthy individuals and most commonly targets the central nervous system [127]. In *T. gondii*-infected astroglia, the PPARγ agonist rosiglitazone reduced neuroinflammation, whereas the PPARγ antagonist GW9662 increased levels of matrix metalloprotease (MMP)-2, MMP-9, and inflammatory mediators. These findings suggested that PPARγ signaling protects against *T. gondii* infection [128]. Proteomic analysis showed that the hepatic protein responses to *T. gondii* infection modulated the PPAR signaling pathways to dysregulate further liver lipid metabolism [129]. However, it remains unclear how *T. gondii*-mediated modulation of PPARγ signaling affects such metabolism and the consequence thereof. 

Sometimes, PPAR signaling negatively affects host defenses against parasitic infections, particularly when M2 macrophage responses are associated with disease progression. During infection of Balb/c mice and hamsters with *Leishmania donovani*, a causative agent of visceral leishmaniasis, the mRNA expression levels of IL-4- and IL-10-driven markers increased significantly [130]. Although any IL-4-related PPARγ function remains unclear, the parasitic load correlated with the effects of IL-10 on the hamster spleen [130]. Schistosomiasis (bilharzia), caused by parasitic flatworms of the genus *Schistosoma*, is associated with inflammatory responses of the intestinal, hepato-splenic, and urogenital systems [131,132]. The Sm16/SPO-1/SmSLP protein from *S. mansoni* may allow the parasite to escape the actions of the innate immune pathway and cellular metabolism, at least partly via a PPAR-dependent pathway [133]. The *Trypanosoma cruzi* protozoan causes Chagas disease, a neglected but chronic tropical infection of great concern in Latin America [134]. During *T. cruzi* infection, both PPARα and PPARγ agonists appear to be involved in macrophage polarization from M1 to M2 types, thereby suppressing inflammation but increasing phagocytosis and macrophage parasitic loads [135]. Thus, PPAR functions may vary by parasite and experimental model. Future studies must explore whether PPARs trigger host defenses or immune evasion during parasitic infections. 

Several PPAR ligands may serve as useful adjunct therapies for Chagas disease, although more preclinical and clinical data are required. The new PPARγ ligand HP24, a pyridinecarboxylic acid derivative, evidenced anti-inflammatory and pro-angiogenic activities and might serve as an adjunct therapy for Chagas disease [136]. 15-deoxy-Δ^12,14^ prostaglandin J_2_ (15dPGJ2), a natural PPARγ agonist, reduced liver inflammation and fibrosis during *T. cruzi* infection [137]. However, the use of PPAR agonists/antagonists should be considered in the context of in vivo PPAR expression levels during certain parasitic infections. For example, acute *T. cruzi* mouse infections trigger significant adipose tissue loss and dysregulation of lipolytic and lipogenic enzymes, which are associated with decreased adipocyte PPAR-γ levels in vivo [138]. Given the robust PPARγ inhibition in this mouse model, PPARγ agonists were minimally effective. However, certain parasites do not specifically affect the host responses depending on PPAR down- or upregulation in target tissues or cells. After infection with the intestinal parasite *Giardia muris*, rapid PPARα induction did not affect the protective or pathological immune responses; PPARα-deficient mice cleared the parasite as did wild-type controls [139]. It is important to explore whether aberrant PPAR expression induced by different parasites improves disease status or, rather, enhances dysfunctional inflammation and infection progression. Table 3 summarizes the roles of PPAR agonists/antagonists in parasitic infections.

## 6. Future Perspectives

PPARs play a wide range of roles across host metabolism, inflammation, and immune responses. Recent studies indicate that PPARs modulate the host responses to infections, such as infectious agent clearance and inflammation. Several PPAR ligands have been utilized in infection models and their functions have been investigated. However, there are no clinical trials of well-known, licensed metabolic medicines utilizing PPAR pathways for infectious diseases. PPAR-based future drugs may serve as adjuvants or components of combination therapies against infections. Understanding the fundamental processes of PPAR-mediated host immune regulation is necessary to develop the most effective treatment approaches for infectious diseases. Future research may also benefit from developing synthetic ligands that preferentially target the specific PPAR isoform implicated in immune response modification.

## 7. Conclusions

Accumulating evidence suggests that PPARs are involved in the host responses to infections caused by bacteria, viruses, and parasites. However, the molecular mechanisms by which PPARs modulate disease progression or protective responses remain unknown. It is essential to further explore the PPAR functions and mechanisms involved in pathogen survival, the pathological responses during different stages of infection, and the associated modulation of the distinct types of infection-associated acute and chronic inflammation. Apart from shaping the inflammatory and metabolic responses during infections, PPARs may impact disease outcomes. The PPAR signaling pathways exert potent immunomodulatory effects; pathway activation or suppression may usefully treat infectious diseases. Infectious pathogens modulate the individual and collaborative activities of PPAR(s) during infection. We speculate that aberrant PPAR expression by various parasites may contribute to inflammation-related dysfunction. It is essential to better understand the possible clinical effects of PPAR-based therapeutics in patients with various infectious diseases.

## Figures and Tables

**Figure 1 cells-12-00650-f001:**
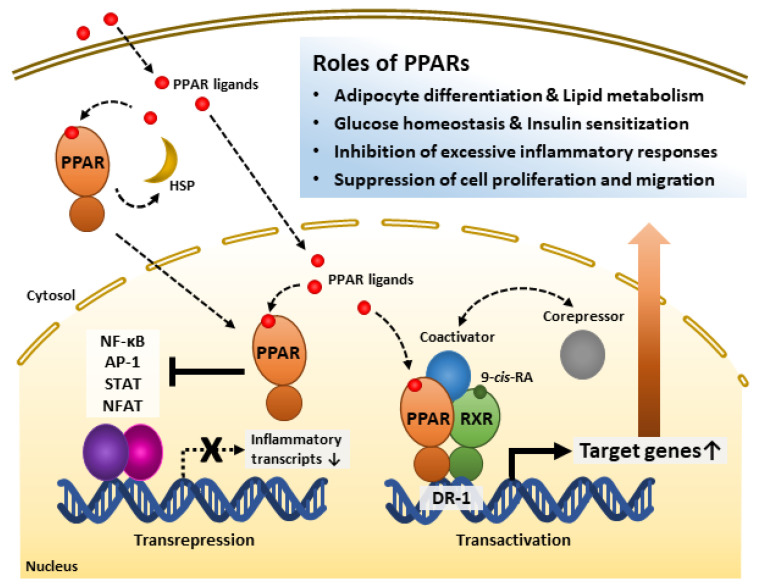
Roles of peroxisome proliferator-activated receptors (PPARs) and their regulatory mechanisms. PPAR ligands bind to the PPAR ligand-binding domain and activate receptors. PPARs interact with heat shock protein (HSP) in the cytosol. PPARs inhibit inflammation-related gene transcription by interfering with transcription factors such as NF-κB, AP-1, STAT, and NFAT. PPARs form heterodimers with Retinoid X receptor (RXR), a receptor of 9-*cis*-retinoic acid (9-*cis*-RA), and bind to direct repeat 1 (DR-1), a peroxisome-proliferator-responsive element. The PPAR-RXR heterodimer complex and co-repressors represses target gene transcription. However, the complex with co-activators promotes target gene transcription. Through these mechanisms, PPARs play significant roles in energy metabolism, inflammatory modulation, and the cell cycle. AP-1, Activator protein 1; NF-κB, Nuclear factor kappa-light-chain-enhancer of activated B cells; NFAT, Nuclear factor of activated T cells; STAT, Signal transducer and activator of transcription.

**Table 1 cells-12-00650-t001:** Studies on PPARs and their ligands during viral infections.

Pathogen	Study Model	Intervention	PPAR Status	Mechanism	Ref.
IAV, RSV	AMs, mice	*Pparg^ΔLyz2^* mice	↓	Regulation of PPARγ through STAT1 activation following IFN signaling	[76]
IAV	AMs, human lung macrophages, mice	*Pparg^ΔLyz2^* mice, Bleomycin	↓	Increased influenza-induced pulmonary collagen deposition in PPARγ-deficient mice	[78]
IAV	A549 cells, mice	Emodin and its analogs	↑	Activation of PPARα/γ and AMPK, decreased fatty acid biosynthesis and increased ATP level	[79]
MERS-CoV	THP-1 cells, primary human monocytes	siRNAs	↑	MERS-CoV S glycoprotein interaction with DPP4 leading to IRAK-M and PPARγ expression	[87]
CHIKV	Vero cells, RAW264.7 cells	Telmisatran, PPAR-γ antagonist GW9662	↓	Activation of PPAR-γ and inhibition of AT1 by telmisartan	[88]
HIV	Primary rat astrocytes, microglia, rats	gp120_ADA_, Rosiglitazone, Pioglitazone	↓	Induction of inflammatory response and decrease in GLT-1 expression in the brain by gp120	[89]
HBV	HepG2.2.15, Huh7, HepG2-NTCP ells	OS_128167, overexpression and downregulation studies, HBV transgenic mice	-	Activation of HBV core promoter by SIRT6 through upregulation of PPARα	[90]
HCV	Huh7.5 cells	Calciterol, Linoleic acid, Ly171883, Wy14643	-	Activation of VDR but inhibition of PPARα/β/γ by calcitriol	[91]

Abbreviations: AMPK, AMP-activated protein kinase; AMs, Alveolar macrophages; AT1, Angiotensin 1; CHIKV, Chikungunya virus; DPP4, Dipeptidyl-peptidase 4; GLT-1, Glutamate transporter 1; HBV, Hepatitis B virus; HCV, Hepatitis C virus; HIV, human immunodeficiency virus; IAV, Influenza A virus; IFN, interferon; IRAK-M, Interleukin-1 receptor-associated kinase 3; MERS-CoV, Middle east respiratory syndrome corona virus; RSV, Respiratory syncytial virus; SIRT6, Sirtuin 6; STAT1, Signal transducer and activator of transcription 1; VDR; Vitamin D receptor; ↑, increase/activation; ↓, decrease/inhibition; -, not reported.

**Table 2 cells-12-00650-t002:** Roles of PPAR agonists/antagonists in bacterial infections.

Pathogen	Drug/Reagent	Function	Study Model	Mechanism of Action	Ref.
*Escherichia coli*	5-aminosalicylic acid	PPARγ agonist	DSS-induced murine colitis model, *Pparg*-deficient mice, CaCo-2 cells	Amelioration of a respiration-dependent luminal expansion of *E. coli*	[104]
*Klebsiella pneumoniae*	Pioglitazone	PPARγ agonist	In vivo mouse model	Reduction of cytokines and myeloperoxidase levels in the lungs	[105]
*Pseudomonas aeruginosa*	Pioglitazone	PPARγ agonist	In vivo mouse model	Increased pro-inflammatory cytokines with enhanced expression of genes involved in glycolysis	[107]
*Chlamydia pneumoniae*	Rosiglitazone	PPARγ agonist	THP-1 macrophages, HEp-2 cells	Regulation of Cpn induced macrophage-derived foam cell formation by upregulating SR-A1 an ACAT1, and downregulating ABCA1/G1 expression via PPARα/γ signaling	[108]
	Fenofibrate	PPARα agonist			
	GW9662	PPARγ antagonist			
	MK886	PPARα antagonist			
*Citrobacter rodentium*	PAR5359	PPARα/γ-dual-agonist	*Citrobacter rodentium*- and DSS-induced murine colitis model, IBD patient-derived PBMCs	Enhanced bacterial clearance, controlled production of ROS and cytokines, anti-inflammatory/healing	[109]
*Rickettsia conorii*	GW9662	PPARγ antagonist	THP-1 macrophages	Increased intracellular survival of bacteria	[112]

Abbreviations: ABCA1/G1, ATP binding cassette transporters A1/G1; ACAT1, acyl-coenzyme A: cholesterol acyltransferase 1; Cpn, *Chlamydia pneumonia*; DSS, Dextran sulfate sodium; IBD, Inflammatory bowel disease; PBMCs, Peripheral blood mononuclear cells; ROS, Reactive oxygen species; SR-A1, scavenger receptor A1.

**Table 3 cells-12-00650-t003:** Roles of PPAR agonists/antagonists in parasitic infections.

Pathogen	Drug/Reagent	Function	Study Model	Mechanism of Action	Ref.
*Angiostrongylus cantonensis*	GW9662	PPARγ antagonist	Mouse model of angiostrongyliasis	NF-κB activation and increase in inflammation and BBB permeability	[124]
*Plasmodium falciparum*	Dimethyl fumarate	-	Cerebral cortex derived HBMVECs	Upregulation of PPAR pathway, NRF2-mediated oxidative stress responses, ErbB4 signaling to downregulate the neuroinflammation	[125]
*Toxoplasma gondii*	Rosiglitazone	PPARγ agonist	SVG p12 cells, Hs68 cells	Decreased expression of MMP-2, MMP-9, COX-2, PGE2, iNOS and NO	[128]
	GW9662	PPARγ antagonist		Increased expression of MMP-2, MMP-9, COX-2, PGE2, iNOS and NO	
*Trypanosoma cruzi*	HP24	pyridinecarboxylic acid derivative	In vivo mice infection, mouse peritoneal macrophages	Induction of PI3K/Akt/mTOR signaling (pro-angiogenic), inhibition of NF-κB signaling (anti-inflammatory)	[136]
	15-deoxy-D12,14 prostaglandin J2	PPARγ agonist	In vivo mice infection	Reduction of liver inflammatory infiltrates, pro-inflammatory enzymes and cytokine expression through inhibition of NF-kB signaling, No change in parasitic load	[137]

Abbreviations: BBB, Blood-brain barrier; COX-2, Cyclooxygenase-2; ErbB4, Erb-b2 receptor tyrosine kinase 4; HP24, 1-methyl-3-hydroxy-4-pyridinecarboxylic acid derivative 24; iNOS, Inducible nitric oxide synthase; MMP, Matrix metalloproteinase; mTOR, Mammalian target of rapamycin; NF-kB, Nuclear factor-κB; NO, Nitric oxide; NRF2, Nuclear factor E2-related factor 2; PGE2, Prostaglandin E2; PI3K, Phosphoinositide 3-kinase.

## Data Availability

Not applicable.

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
