# Peer review of "Peroxisome Proliferator-Activated Receptor-Targeted Therapies: Challenges upon Infectious Diseases"

_cells, 2023, doi:10.3390/cells12040650_

Round 1

Reviewer 1 Report

I found this study very interesting . We generallyknow the roles of peroxisome proliferator-activated receptors (PPARs) in cancer and diabetes. However, to discuss their efficacy in infectious disorders is very different and reasonable. 

Minor comments: Authors should add future directions and remarks just before the conclusion part.

Author Response

RESPONSES

Thank you for your kind review and comment. We added the new chapter of future directions about PPARs on infectious disease study before the conclusion part in lines 430-441.

Lines 430-441:

  1. Future perspectives

PPARs play a wide range of roles across host metabolism, inflammation, and im-mune responses. Recent studies indicate that PPARs modulate the host response to in-fections, such as infectious agent clearance and inflammation. Several PPAR ligands have been utilized in infection models, and their functions have been investigated. However, there are no clinical trials of well-known, licensed metabolic medicines utilizing PPAR pathways for infectious diseases. PPAR-based future drugs may serve as adjuvants or components of combination therapies against infections. Understanding the fundamental processes of PPAR-mediated host immune regulation is necessary to develop the most effective treatment approaches for infectious diseases. Future research may also benefit from developing synthetic ligands that preferentially target the specific PPAR isoform implicated in immune response modification.

Reviewer 2 Report

This review by Kim et al. is timely and original. I think it will be very useful to the people interested in PPAR's biology. 

I have one minor concern that I think the authors may wish to consider. The authors make a very thorough review about the evidences linking direct control of important genes involved in immune responses. They do not review much of the indirect effects related to the non-autonomous metabolic response to PPAR activation.

In my opinion, it would be nice to refer to the major metabolic roles of PPARalpha. The best known example relates to hepatic PPARalpha. In the liver, PPARalpha is highly expressed and determines the response to fasting.

Kersten et al, https://doi.org/10.1172/jci6223

Hepatocyte PPARalpha controls catabolism of fatty acids, ketogenesis and FGF21 production.

Montagner et al, https://doi.org/10.1136/gutjnl-2015-310798

This genomic response relies on adipose to liver cross-talk that determines ketone bodies production as well as FGF21

Fougerat et al, https://doi.org/10.1016/j.celrep.2022.110910

Ketone bodies are critical in the response to either microbial or viral infections

Wang et al, https://doi.org/10.1016/j.cell.2016.07.026

Hepatocyte FGF21 is also key in bacterial infection

Huen et al,  https://doi.org/10.1084/jem.20202151

And hepatocyte PPARalpha is critical in the response to sepsis

Paumelle et al,  https://doi.org/10.1016/j.jhep.2018.12.037

This is a very useful review regarding the different roles of the 3 PPAR isotypes in infectious challenges. The description of these roles primarily focus on the regulation of inflammatory response. The review is very well done and useful. However, the authors could consider to include the regulation of hepatic ketogenesis and FGF21 production as a significant host metabolic response to infection. This is a clearly established PPAR(alpha)-dependent response to fasting induced by infection (sickness behavior) that is not currently mentioned in this review. I suggested a few publications to the authors regarding this metabolic response to infection and the evidences that it involves hepatocytic PPARalpha activity. But this is not a mandatory change to be if the authors are not convinced it is relevant. That is why I said it is a minor change.

Author Response

RESPONSES

Thank you for your valuable comments. We agree with the importance of non-autonomous metabolic responses to PPAR activation in infectious diseases. Therefore, we mentioned PPAR alpha-dependent hepatic responses in infection status in lines 142-152.

Lines 142-152:

In the context of infection, PPARα has been shown to play an essential role in the hepatic metabolic response to infection. During an infectious challenge, the liver coordi-nates several metabolic changes to support the host defense response, including the mo-bilization of energy stores, the production of acute-phase proteins, and the synthesis of new metabolites. Activation of PPARα in the liver leads to the upregulation of genes involved in fatty acid oxidation and ketogenesis with fibroblast growth factor 21 (FGF21) production [58]. FGF21 is a hormone produced by the liver that has been shown to pro-mote ketogenesis and reduces glucose utilization [59,60]. The ketogenesis regulation of PPARα with FGF21 is essential for reacting to microbial or viral sepsis [61-63]. In con-clusion, the hepatic PPARα metabolic response to infection is crucial to the host defense response.
